# Antimicrobial resistance in topical treatments for microbial keratitis: protocol for a systematic review and meta-analysis

Stephen Tuft ,[1,2] Jennifer Evans,[3] Iris Gordon,[4] Astrid Leck,[4] Neil Stone,[5] Timothy Neal,[6] David Macleod,[7] Stephen Kaye,[8] Matthew J Burton[3]

For numbered affiliations see end of article.

**Correspondence to**
Dr Stephen Tuft;
s.tuft@ucl.ac.uk

## ABSTRACT

**Introduction** There is evidence for increased resistance against the antimicrobials used to treat keratitis. This review aims to provide global and regional prevalence estimates of antimicrobial resistance in corneal isolates and the range of minimum inhibitory concentrations (MIC) with their associated resistance breakpoints.

**Methods and analysis** We report this protocol following Preferred Reporting Items for Systematic Review and Meta-Analyses Protocols guidelines. We will conduct an electronic bibliographic search in MEDLINE, EMBASE, Web of Science and the Cochrane Library. Eligible studies will report in any language data for the resistance or MIC for antimicrobials against bacterial, fungal or amoebic organisms isolated from suspected microbial keratitis. Studies that only report on viral keratitis will not be included. There will be no time restrictions on the date of publication. Screening for eligible studies, assessment of risk of bias and data extraction will be conducted by two reviewers independently, using predefined inclusion criteria and prepiloted data extraction forms. We will resolve disagreements between the reviewers by discussion and, if required, a third (senior) reviewer will arbitrate. We will assess the risk of bias using a tool validated in prevalence studies. The certainty of the evidence will be assessed using the Grades of Recommendation, Assessment, Development and Evaluation approach. Pooled proportion estimates will be calculated using a random-effects model. Heterogeneity will be assessed using the $I^2$ statistic. We will explore differences between Global Burden of Disease regions and temporal trends.

**Ethics approval and dissemination** Ethics approval is not required as this is a protocol for a systematic review of published data. The findings of this review will be published in an open-access, peer-reviewed journal.

**PROSPERO registration number** CRD42023331126.

## STRENGTHS AND LIMITATIONS OF THIS STUDY

⇒ This systematic review protocol follows the Preferred Reporting Items for Systematic Review and Meta-Analyses Protocols guidelines.
⇒ This systematic review addresses a gap in the current evidence by estimating the prevalence of antimicrobial resistance and minimum inhibitory concentrations (MICs) by Global Burden of Disease region in common corneal pathogens.
⇒ There may be significant clinical, methodological and statistical heterogeneity in reporting prevalence data between different populations.
⇒ A potential limitation might be the lack of published literature on MICs and the method of extrapolation of this data to the definition of clinical resistance.
⇒ Further potential limitations are the application of clinical thresholds for sampling, and sampling of cases in a hospital setting that may not be representative of the full spectrum of microbial keratitis.

## INTRODUCTION

Microbial keratitis (MK) is a common corneal infection treated with topical antimicrobials. The causative organisms include bacteria, fungi, protozoa and viruses, either singly or in combination. The proportions of these pathogens (viruses, bacteria, parasites, fungi) vary widely between geographic regions, with fungal keratitis particularly prevalent in low-income and middle-income countries and equatorial areas.[1] Published figures for the incidence of MK include 113 per 100 000 person-years in South India[1–3] and 799 per 100 000 person-years from Nepal.[4] A population-based study in China estimated the prevalence of past or active microbial keratitis to be 192 (95% CI 171 to 213) per 100 000, with a prevalence of presumed viral keratitis of 110, bacterial keratitis 75 and fungal keratitis 7 per 100 000.[2] The incidence figures for MK reported from high-income countries since 1995 are much lower with estimates of 4.5–37.7 cases per 100 000 person-years in the USA, UK, Australia and Taiwan,[3] which follows a marked increase in cases from the 1970s onward associated with the widespread introduction of contact lens wear.[4]

The relative frequency and distribution of the potential bacterial pathogens isolated from corneal ulcers are usually derived from

retrospective data rather than prospective surveys. Ideally, all ulcers should be sampled, as defining a threshold ulcer size to justify corneal culture will introduce case selection bias that could affect the spectrum of isolates.[5 6] The proportions vary widely between reports and according to the definition used for a significant isolate. In a recent meta-analysis of 38 studies, the most common isolates were *Staphylococcus* spp (including *Staphylococcus aureus* and coagulase-negative staphylococci) (41.4%, 95% confidence limits 36.2%–46.7%), *Pseudomonas* spp (17.0%, 13.9%–20.7%), *Streptococcus* spp (13.1%, 10.9%–15.7%), *Corynebacterium* spp (6.6%, 5.3%–8.3%) and *Moraxella* spp (4.1%, 3.1%–5.4%).[7] *Streptococcus pneumoniae* and *Nocardia* spp. are more frequently reported in series from South India.[8–10] Differences in environmental temperature, humidity, occupation, associated viral disease and malnutrition affect the local prevalence pattern.[11] In low-income countries, agricultural trauma is the major risk for infection due to filamentous fungi and *Acanthamoeba* spp., as opposed to contact lens wear in high-income regions.[12]

The goal of antimicrobial susceptibility testing is to reliably produce data that may be used to guide patient therapy, inform epidemiological studies, and track rates of antimicrobial resistance (AMR). AMR is measured by the response of a pathogen cultured in vitro in the presence of increasing concentrations of an antimicrobial agent. The critical measurement for antimicrobial susceptibility testing is the minimum inhibitory concentration (MIC) - the concentration of the antimicrobial that inhibits overnight growth of the microbe cultured from the patient. This MIC is then referenced to a breakpoint, a chosen antimicrobial concentration (mg/L) that defines the likelihood of clinical success for that agent against the microbe. The breakpoint is a tool to link a MIC value generated by in vitro testing to the most likely in vivo response of that isolate to an achievable drug concentration. If there is an established breakpoint, the MIC can then be used to differentiate isolates in which there is a high likelihood of treatment success from those in which treatment is more likely to fail. The breakpoint is determined from the MIC distribution of a well-characterised wild-type population, the pharmacokinetics of the antimicrobial in vivo, host status, and clinical trials, to identify the lowest MIC to give a good clinical outcome. Ideally, the MIC should be at the lower end of the achievable tissue concentration. However, in vitro resistance does not necessarily equate to treatment failure. The clinical breakpoints of topically applied antimicrobials are unknown, and values for the topical treatment of corneal infection are based on achievable and safe serum concentrations that may not be relevant for topically applied antimicrobials. Suggested ophthalmic breakpoints for bacteria and fungi are provided by regulatory authorities such as EUCAST or CSLI.[13 14] For microbial keratitis, it is difficult to establish a breakpoint without outcome data, so defining the epidemiological cut-off value (ECV or ECOFF) is an alternative. The ECOFF is not equivalent

to a breakpoint but is the MIC that separates isolates into those with or without acquired or mutational resistance by defining the upper limit of the wild-type MIC distribution. Again, it does not predict clinical success but indicates whether a specific isolate carries resistance to an antimicrobial that would otherwise be active against that species.

AMR is an intrinsic or acquired characteristic encoded by genes that can be transferred between bacteria.[15] It is a global phenomenon that has made some systemic bacterial infections challenging to treat.[16–19] An increase in the prevalence of bacterial resistance to fluoroquinolones, beta-lactams and aminoglycosides has been reported in ophthalmic isolates.[8 18 20–24] Although *P. aeruginosa* susceptibility to either ciprofloxacin or moxifloxacin for corneal isolates is still approximately 80% worldwide,[8 23–26] several studies have highlighted the presence of high proportions of AMR bacteria in ocular infections, particularly in the US,[27] China,[28] and India.[29] A particularly sharp increase in resistance of *P. aeruginosa* to moxifloxacin occurred between 2007 and 2009 in southern India, from 19% (95% CI 5.4% to 41.9%) to 52% (95% CI 29.8% to 74.3%) (p=0.024).[30] Two separate 20year reviews from the US also found an increase in methicillin-resistant *S. aureus* keratitis from 1993 to 2015.[23 31] Fluoroquinolone-resistance, methicillin-resistance and multidrug-resistant (MDR) bacteria, defined as acquired resistance to at least one antimicrobial agent in three or more antimicrobial classes, has also been reported in isolates from bacterial keratitis.[25 32–35] *P. aeruginosa* can show MDR,[36 37] with MDR *P. aeruginosa* emerging as a cause of bacterial keratitis in South Asia.[38 39] In South China, an increase in AMR in Gram-positive cocci was reported between 2010 to 2018, while the susceptibility of Gram-negative bacilli to fluoroquinolones and aminoglycosides was stable.[28]

Evidence for a change in AMR to the agents used to treat fungal or acanthamoeba keratitis is less certain. Susceptibility testing for antifungal agents is usually only performed in large hospitals or national reference laboratories,[40] and susceptibility testing for antiamoebic agents is rarely performed. The prevalence of acquired resistance to antifungal agents is generally lower than antibiotic resistance, but the treatment options are more limited.[41 42] Most data on the prevalence of resistance to antifungal agents is derived from testing isolates from systemic infections, with relatively few studies of corneal isolates.[43–46] Methods to assess fungal resistance include broth dilution, disc dilution, antifungal impregnated gradient strips and agar screening as described by EUCAST or the CSLI. Resistance may also be intrinsic (eg, resistance of *Candida krusei* to fluconazole) or an acquired trait following prolonged drug exposure. If resistance is uncommon (eg, *Aspergillus fumigatus* to voriconazole), empiric treatment can be determined by accurately identifying the pathogen without routine susceptibility testing. In addition, if there is no interpretive data (breakpoint or ECOFF), the MIC alone may be of limited clinical value. For isolates from systemic

infection, there is a trend toward infection with non-*albicans* species of *Candida* and an increase in antifungal resistance,[47] and acquired resistance has emerged in species such as *Candida glabrata* and *Candida auris* (rarely reported as an ocular pathogen). There has also been an increase in the resistance of *Aspergillus fumigatus* to azoles, especially in Europe,[48] linked to the use of azoles in agriculture.[49] However, polyenes (natamycin, amphotericin B) are preferred for empiric therapy of fungal keratitis rather than azoles.[50 51] Therapeutic guidelines for fungal keratitis have largely been developed from randomised controlled trials (RCTs) performed in South Asia.[50–52] Importantly, the vast majority of isolates from these studies were filamentous fungi, although yeast contributes approximately 25% of isolates in case series from temperate regions.[53] Comparisons of antifungal resistance between regions for isolates from keratitis have not been performed.

Speciation of isolates of *Acanthamoeba* spp. or susceptibility testing against biocides is not usually performed or discussed by the EUCAST or CSLI. There is a lack of consensus on the optimum protocol for testing trophozoites or cysts against antiseptics (polyhexanide, chlorhexidine).[54] When it is done, susceptibility testing is performed on cysts because most amoebicides readily kill trophozoites. The usual test is based on preventing excystment after culture, or assessing cyst viability by staining with trypan blue, following exposure to serial dilutions for one to 7 days in vitro.[55 56] However, this protocol is primarily used for the assessment of potential amoebicidal drugs rather than for clinical management or audit of resistance. A few publications that have used these methods show a poor correlation between in vitro susceptibility and clinical eradication of acanthamoeba from the cornea.[57 58]

A recent systematic review and meta-analysis estimated differences in the spectrum of bacterial isolates between regions, the overall prevalence of bacterial AMR, and changes over time.[7] However, there have been no reviews of resistance in either fungal or amoebic keratitis. The review by Zhang *et al* was limited to citations in English published after 2000, with an emphasis on reporting positive culture rates and the spectrum of bacterial isolates globally, with no regional data.[7] We will extend the study to all microbes, and include all publications with no limit to time or language. We will report differences in resistance between Global Burden of Disease (GBD) regions. Because ophthalmic breakpoints are unknown, we will also record the MIC data where this is available. This is because reporting the prevalence of resistance does not identify the primary measurement, which is the MIC. In addition, changes in MIC can occur before the effect of this is evident as a change in resistance values (MIC drift). Establishing the prevalence of antimicrobial resistance may focus attention on the magnitude of the problem and serve as a benchmark for future publications.

## Objectives

- ► To report the proportion of the common pathogens isolated from microbial keratitis that are resistant to the antimicrobials commonly used as empiric therapy and report the distribution of their MICs.
- ► To explore how the proportion of AMR and the distribution of MICs varies by region.
- ► To explore if the proportion of AMR and the distribution of MICs in common pathogens has changed over time.

## METHODS AND ANALYSIS
### Protocol and registration
We report this protocol following the Preferred Reporting Items for Systematic Reviews and Meta-Analyses Protocols guidelines.[59] The protocol has been registered in PROSPERO (registration number CRD42023331126). Any protocol amendments will be documented, including the date of amendment and rationale for the change in the final review report.

### Types of studies
We will include published prospective and retrospective cross-sectional studies that have identified the genus and species of isolates from corneal cultures and recorded the susceptibility or MIC of each isolate to eye-appropriate topical antimicrobials. We define eye-appropriate antimicrobials as agents that are either commercially available or specifically compounded for topical ophthalmic use. We will also consider the baseline findings (pre-randomisation) of people recruited in clinical trials for treatment or cohort studies. We will only include peer-reviewed articles published in any language. Articles in a language other than English will be screened with an internet-based translation service (eg, Google Translate). If they appear relevant a professional translation will be obtained. We will not include grey literature or conference abstracts. We will consider publications from all years and geographical regions.

### Types of participants, eligibility and setting
Inclusion criteria will be studies that report the investigation results for suspected microbial keratitis, including bacterial, fungal and acanthamoeba keratitis. We will exclude studies that only report on viral isolates or in vitro resistance to antivirals. There will be no age restrictions. Settings will include primary care as well as secondary and tertiary referral centres. Studies must report the results of an investigation, including the identification of the organisms grown in culture (genus and species) and their anticipated susceptibility or resistance to antimicrobials, with the measured MIC if this is available. The genus and species will be recorded. The method of assessment (disc diffusion, Etest, tube dilution) will be recorded. The regulatory body that provided the assessment thresholds or breakpoints will be recorded, such as EUCAST, CSLI, ChiCAST. We will include case series, but not individual case reports, as we require a representative

sample of isolates from a particular setting with information on the proportion with AMR. We will exclude studies where participants are exclusively patients in specific settings (eg, intensive care units), or patients with specific syndromes (eg, Stevens Johnson syndrome) or other disease groups (non-healing ulcers, corneal graft for treatment failure). We will include studies irrespective of pre-treatment of patients. We will group the results as bacteria, fungi or acanthamoeba, and we will pool data from studies within regions. The absolute number of isolates and the proportion of the total isolates will be recorded. Differences between the method of identification (MALDI-TOF vs biochemical assay) for bacteria is not thought to be significant. Results from PCR will not be included as these do not routinely measure AMR. We will not analyse the response to therapy.

## Types of outcome measures

▶ The proportion of the common pathogens isolated from suspected microbial keratitis that are resistant to the antimicrobials commonly used as empiric therapy. Results will be recorded as sensitive (S), sensitive at an increased dosage (I), or resistant (R). For the analysis, the S and I groups will be amalgamated.

▶ The MIC values (mg/L) for the antimicrobials commonly used as empiric therapy for suspected microbial keratitis. Results will be reported as the mode and range.

## Search strategies
### Electronic searches
We will conduct an electronic bibliographic search in MEDLINE, EMBASE, Web of Science, and the Cochrane Library. An expert information specialist developed the search strategies (online supplemental figure 1). Relevant publications will be retrieved manually if electronic access is not available.

### Searching for other resources
We will identify additional studies by searching the reference lists of relevant publications identified through electronic searches and prior review articles on this topic. We will consult trial registries such as WHO ICTRP and ClinicalTrials.gov to identify studies indexed in the databases.

### Selection of studies
Two review authors will independently screen search results based on title and abstract and will remove reports that do not fall into the scope of this review. We will resolve disagreements by discussion and consultation with another author as needed. We will acquire the full text of all publications appearing to meet the criteria for inclusion in this review. Two review authors will independently screen the full-text reports using the inclusion criteria listed above. They will discuss any disagreements, and a third review author will arbitrate if they cannot resolve them. Screening of search results will be conducted using Covidence systematic review software (Veritas Health

Innovation, Melbourne, Australia; available at www.covidence.org).

## Data extraction and management
Data extraction will be done in Covidence using a customised data extraction template which will be pre-piloted on five studies (table 1). Two review authors will independently extract the following data from each study: study design, participant characteristics, study population and size, study setting, study dates, diagnostic and ascertainment methods, details of pathogen and antimicrobials, number of isolates, the proportion of isolates reported resistant (R) for each antimicrobial, and the MIC values of these antimicrobials. We will document the method used for speciation of the organism, how the MIC was determined, and the reference guidelines for determining breakpoint thresholds. Disagreements will be resolved by discussion and consultation with another author as needed; a third review author will arbitrate if they cannot resolve them.

AMR will be recorded separately for bacteria, fungi and acanthamoeba. Within each phylum, the AMR will be reported for each class of antimicrobial against each genus of organism.

▶ For bacterial isolates, the genus and species of all isolates will be recorded, with a secondary grouping of pathogens into classes. The groups will be: *Enterobacterales, Pseudomonas* spp.*, Staphylococcus* spp.*, methicillin-resistant S. aureus* (MRSA), *coagulase-negative staphylococci (CNS)*, *Streptococcus pneumoniae,* viridans group streptococci (*VGS*), *beta-haemolytic streptococci, Haemophilus* spp.*, Neisseria* spp. The antimicrobial groups will be quinolones (second and subsequent generations), beta-lactams (early and late cephalosporins), aminoglycosides, chloramphenicol, glycopeptides and antiseptics.

▶ For fungal isolates, the genus and species will be recorded, with a secondary grouping of the pathogens into classes. The groups will be yeast and filamentous fungi. The antifungal groups will be the polyenes (natamycin, amphotericin), azoles (miconazole, econazole, voriconazole), echinocandins (micafungin) and antiseptics.

▶ For amoeba isolates the genus will be recorded and the species if this has been reported. The anti-amoebic treatments include polyhexanide, chlorhexidine, voriconazole and neomycin.

## Assessment of risk of bias
Two reviewers will independently assess the risk of bias in each included study using the risk of bias tool for prevalence studies developed by Hoy *et al.*[60] This tool covers four domains - selection bias, non-response bias, measurement bias and bias related to analysis. For each of the 10 items included in the assessment, two reviewers will record the rationale for their judgements. The Risk of Bias (online supplemental figure 2) shows the items included in the tool and specific considerations for this

**Table 1** Data extraction domains for included studies

|   | Domain/subdomain | Description |
|---|---|---|
| 1 | *Document characteristics* | |
|   | Title, authors, publication year | Title, authors and year of publication |
|   | Full citation and web link | Citation of publication and PMID |
| 2 | *Study characteristics* | |
|   | Design | Prospective study, retrospective study, RCT, etc. |
|   | Setting | Primary care, hospital eye service, tertiary referral centre, number of participating centres |
|   | Location | City and country of data collection, GBD region |
|   | Data collection | Years when samples collected |
|   | Population | Eligibility criteria, age range, specific risk group (LVC, ICU, SJS) |
|   | Sample size | Number of participants in the study. Eyes included. |
|   | Study objectives | What was the study research question? |
| 3 | *Characteristics* | |
|   | Inclusion criteria | Presumed microbial keratitis, size and position of ulcer |
|   | Exclusion criteria | Case report, selected disease group, for example, corneal perforation |
|   | Definition of significant isolate | Growth on one or more media, supportive investigation (histology, PCR, IVCM) |
|   | Microbial isolate | Bacteria, fungus, acanthamoeba |
|   | Method of identification | eg, MALDI-TOF, phenotypic and biochemical, molecular |
|   | Determination of MIC | Tube dilution, Etest, disc diffusion, agar slope, excystment. |
|   | Reference body for breakpoint | EUCAST, CSLI, ChiCAST |
|   | Bacterial groups | *Enterobacteriales, Pseudomonas* spp, *Staphylococcus* spp, MRSA, *Staphylococcus* (CNS), *Streptococcus pneumonia, Streptococcus* (VTS), beta-haemolytic Streptococcus, *Haemophilus* spp, *Neisseria* spp. |
|   | Sensitivity and MIC of bacteria to antimicrobials (six categories) | Quinolones, beta-lactam, aminoglycoside, glycopeptide, chloramphenicol, antiseptics |
|   | Fungal groups | Yeast, filamentary fungus |
|   | Sensitivity and MIC of fungi to antifungals (four categories) | Polyenes, azoles, echinocandins, antiseptics |
|   | Amoeba isolates | Acanthamoeba with speciation if available |
|   | Sensitivity and MIC of amoeba to amebicides (four categories) | Polyhexanide, chlorhexidine, voriconazole, neomycin |

MIC, minimum inhibitory concentration; EUCAST, European Committee on Antimicrobial Susceptibility Testing; CSLI, Clinical & Laboratory Standards Institute; MRSA, methicillin-resistant Staphylococcus aureus; VTS, viridans-type streptococci; MALDI-TOF, matrix-assisted laser desorption/ionisation time of flight; LVC, laser vision correction; ICU, intensive care unit; SJS, Stevens Johnson syndrome; PCR, polymerase chain reaction; IVCM, in vivo confocal microscopy; GBD, Global Burden of Disease; RCT, randomised control trial; .

review. This risk of bias tool will be piloted on five studies and instructions for reviewers amended in the light of that pilot. Agreement between reviewers will be assessed and any disagreements resolved by discussion. For each study, a judgement of the overall risk of bias (low, moderate, or high) will be made.

### Dealing with missing data

If we do not find all the necessary information in a published study, for articles published in 2010 or later, we will email the corresponding author to solicit further information. If we cannot obtain the necessary information, we will document in the review that we attempted to contact the study authors. We will report and discuss the possible effect of missing data on each study and the overall review and meta-analysis. We will consider the susceptibility of our meta-analysis to the impact of missing data. We will analyse the data that is available rather than imputing missing data.

### Unit of analysis issues

We will document how each study handled eyes - whether one eye or both eyes were included and analysed in the study. We will also document how the study handled multiple isolates per person or eye. Where multiple isolates are identified we will consider each isolate separately and analyse as reported. Where possible, we will extract proportions/confidence intervals adjusted for clustered data or consider adjusting using methods outlined in the Cochrane Handbook.[61]

### Data synthesis and meta-analysis approach

We will perform meta-analyses using a random-effects model in Stata V.17 (StataCorp).[62] We will summarise and

report the pooled proportion of resistance, its 95% CI, and between-study heterogeneity ($I^2$). We will conduct meta-analyses separately for each class of antimicrobial (table 1): fluoroquinolones (second and fourth generation separately), beta-lactam antibiotics (eg, cefuroxime), aminoglycosides (eg, gentamicin, tobramycin), glycopeptide antibiotics (eg, vancomycin) for bacterial; natamycin, voriconazole, amphotericin, (and others) for fungi (filamentous fungi, yeasts); polyhexanide (PHMB) and voriconazole for *Acanthamoeba* spp. If a study reports susceptibility for more than one antimicrobial in a class against the same organism(s), our strategy for the primary analysis will be to select the results for the most frequently used antimicrobial of that class from the whole data set. We will then perform a sensitivity analysis substituting results for the less commonly used antibiotic(s) in that class to check that the results do not change substantially. We will consider amalgamating closely related groups of organisms (table 1) that have similar mechanisms of AMR.

We anticipate heterogeneity in reporting of MIC values and therefore we plan to report these data narratively in structured tables and figures. If comparable figures on MIC in mg/L are available from different studies, we will consider a meta-analysis as specified above.

## Meta-regression

If there are sufficient studies for analysis, we will include the following covariates:

▶ Class of microbe: We intend to present data for the eight most frequent Gram-positive (*S. aureus*, *Streptococcus* spp etc) and Gram-negative (*P. aeruginosa*, etc) isolates.
▶ Region: we will group studies according to the global super-region as defined by the Global Burden of Disease Study.[63]
▶ Decade: we will group studies according to the decade in which they were conducted: Before 2000, 2000–2009, 2010–2019, 2020–2022.
▶ Pre-treatment (none, some, all).
▶ Sex (M:F).
▶ Age (children:adults >16 years).
▶ Study size. We will assess small study effects, one of which may be publication bias, by preparing a funnel plot, which is a scatter plot of effect size vs precision (SE).[64]

## Assessment of heterogeneity

We will assess clinical heterogeneity by comparing key participant characteristics at the study level (eg, age, sex, ocular diagnosis). Methodological heterogeneity will also be considered, including comparing the risk of bias of included studies. We will assess statistical heterogeneity by inspecting forest plots and through the $I^2$ statistic. If high levels of inconsistency are detected, and there are enough studies, we will explore likely sources of this heterogeneity (see section on meta-regression).

## Sensitivity analysis

We will conduct a sensitivity analysis of the primary analysis in which studies are excluded if they are judged to be at high risk of bias.

## Assessing the certainty of the evidence

We will assess and report the overall certainty of evidence from our analyses and critical appraisal using the modified Grades of Recommendation, Assessment, Development and Evaluation (GRADE) tool.[65] We will consider evidence from cross-sectional studies to be high certainty and will downgrade for risk of bias, imprecision, inconsistency, indirectness and publication bias. The GRADE assessment will be done by consensus discussion.

## Patient and public involvement

None.

# ETHICS AND DISSEMINATION

Ethics approval is not required as this is a protocol for a systematic review of published data. Findings will be published in an open-access peer-reviewed journal and presented at national and international meetings. We anticipate that the findings will be of considerable interest to those involved in eye health provision, as well as the general medical, public health, development, and governmental sectors.

**Author affiliations**
[1]Cornea and External Disease Department, Moorfields Eye Hospital NHS Foundation Trust, London, UK
[2]UCL Institute of Ophthalmology, University College London, London, UK
[3]Faculty of Infectious and Tropical Diseases, LSHTM, London, UK
[4]Department of Infectious and Tropical Diseases, LSHTM, London, UK
[5]Department of Clinical Microbiology, University College London Hospitals NHS Foundation Trust, London, UK
[6]Department of Microbiology, Royal Liverpool University Hospital, Liverpool, UK
[7]Department of Medical Statistics, LSHTM, London, UK
[8]Department of Eye and Vision Science, University of Liverpool, Liverpool, UK

**Contributors** ST conceived the original idea. ST and JE wrote the first draft of this protocol, and MJB, TN, DM, SK, IG, AL and NS made further suggestions. ST, JE, DM, NS, TN, AL, SK and MJB participated in the study's design and the inclusion and exclusion criteria setting. IG, JE and ST developed the search strategy. ST, JE, MJB, SK and DM will supervise the data extraction and the overall work. ST and JE are the guarantors. All authors read the final draft and approved the publication of the protocol.

**Funding** MJB, AL and DM and supported by Wellcome Trust (207472/Z17/Z). The funder had no role in the development of the review.

**Competing interests** None declared.

**Patient and public involvement** Patients and/or the public were not involved in the design, or conduct, or reporting, or dissemination plans of this research.

**Patient consent for publication** Not applicable.

**Ethics approval** Not applicable.

**Provenance and peer review** Not commissioned; externally peer reviewed.

**Data availability statement** All data relevant to the study are included in the article or uploaded as supplementary information.

responsibility arising from any reliance placed on the content. Where the content includes any translated material, BMJ does not warrant the accuracy and reliability of the translations (including but not limited to local regulations, clinical guidelines, terminology, drug names and drug dosages), and is not responsible for any error and/or omissions arising from translation and adaptation or otherwise.

**ORCID iD**
Stephen Tuft http://orcid.org/0000-0001-8192-5192

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
