## [Reviewer comments · BMJ Open]

This paper was submitted to a another journal from BMJ Open Ophthalmology but declined for publication following peer review. The authors addressed the reviewers' comments and submitted the revised paper to BMJ Open. The paper was subsequently accepted for publication at BMJ Open.

(This paper received one review from its previous journal and one reviewers agreed to published their review.)

ARTICLE DETAILS

TITLE (PROVISIONAL)	Antimicrobial resistance in topical treatments for microbial keratitis: protocol for a systematic review and meta-analysis
AUTHORS	Tuft, Stephen; Evans, Jennifer; Gordon, Iris; Leck, Astrid; Stone, Neil; Neal, Timothy; Macleod, David; Kaye, Stephen; Burton, Matthew J

VERSION 1 – REVIEW

REVIEWER	Jennifer Rose-Nussbaumer Stanford University, Ophthalmology
REVIEW RETURNED	20 Jul 2022

GENERAL COMMENTS	This protocol paper describes a meta-analysis of antimicrobial resistance patterns. The meta-analysis itself will be of great interest. Although this protocol paper is well written, I am not sure that a protocol paper describing a meta-analysis is warranted. Major Comments: Will you evaluate criteria for defining culture positive results, there be any criteria for defining for culture positive (i.e. growth on two media for coag neg staph, etc) Will you assess primary ophthalmology settings separately from tertiary ophthalmology settings since they are likely to have different organism and anti-microbial resistance patterns? How will you read articles that are not in a language that the authors understand? Will you assess how in vitro MIC correlates to response to therapy? Minor Comments: Line 44 is confusing, you are reporting on bacterial, fungal and acanthamoeba keratitis. Presumably you will not include studies of viral keratitis at all?
--

VERSION 1 – AUTHOR RESPONSE

Reviewer(s)' Comments to Author:

Reviewer: 1

Comments to the Author

This protocol paper describes a meta-analysis of antimicrobial resistance patterns. The meta-analysis itself will be of great interest. Although this protocol paper is well written, I am not sure that a protocol paper describing a meta-analysis is warranted.

Response: We believe that it is best practice to publish the protocol before a systematic review and metanalysis is performed. This is an opportunity for the paper to be reviewed by a methodologist. This may identify areas where the robustness of the review could be improved.

Major Comments:

Will you evaluate criteria for defining culture positive results, there be any criteria for defining for culture positive (i.e., growth on two media for coag neg staph, etc).

Response: The criteria for distinguishing a positive culture (pathogen) from a contaminant are weak in the context of microbial keratitis. It has been argued that for suspected corneal infection all microbes growing within the areas of inoculation should be considered potential pathogens (PMID 35102245). No bacterial species (e.g., coagulase negative Staphylococci) will not be excluded from our analysis. We will record any definitions of a significant isolate provided in the publications, but this will not be the basis for exclusion from the analysis. The definition used to define an infection is now included in the updated Table 1. "Data extraction domains for included studies".

Will you assess primary ophthalmology settings separately from tertiary ophthalmology settings since they are likely to have different organism and anti-microbial resistance patterns?

Response: Yes, the effect (if any) of the location of the study will be considered. This point is included in Table 1. "Data extraction domains for included studies".

How will you read articles that are not in a language that the authors understand?

Response: Articles in a language other than English will be screened with an internet-based translation service (e.g., Google Translate). If they appear relevant a professional translation will be obtained. The text has not been modified.

Will you assess how in vitro MIC correlates to response to therapy?

Response: No, the response to therapy is not part of this analysis. For clarity, we have added the following text to page 9:

"We will not analyse the response to therapy."

Minor Comments:

Line 44 is confusing, you are reporting on bacterial, fungal and acanthamoeba keratitis. Presumably you will not include studies of viral keratitis at all?

Response: That is correct. We will not include cases of suspected viral keratitis or data on in-vitro response to antivirals. The text has been modified for clarity (page 9:line 44):

“We will exclude cases or studies that only report on viral isolates or in vitro resistance to antivirals.”

VERSION 2 – REVIEW

REVIEWER	Atalay, Eray Eskisehir Osmangazi University, Turkey
REVIEW RETURNED	02-Jan-2023

GENERAL COMMENTS	This is a protocol paper describing a well-contemplated meta-analysis on the AMR in topical treatments for microbial keratitis. Considering the scope and breadth of the analysis, it will provide invaluable information both for clinicians and researchers alike. My comments can be found below. Page 4 line 53: Please write the year range in full e.g., 2007 – 2009 Page 9 line 5: Please change the wording “recorded” to “record”. What do the authors mean by eye-appropriate topical antimicrobials. Please clarify. Page 9 line 22: Although it is understood in the preceding paragraph, please clarify whether “case series” or “case reports” will be included/excluded. Please also clarify whether studies that report keratitis of viral origin (with/without antiviral susceptibility testing) will be included/excluded. Although this has been partially addressed in response to a previous comment, the cited sentence creates confusion as to whether "cases" or "case series" of other isolates will be included or not. Please also report the target isolate groups (bacterial, fungal and amebic) in the abstract. Page 9 line 42: please write acronyms (ICU, SJS) in full at first use. Page 11 line 49: there is emerging data that CNS can also demonstrate methicillin resistance. How will these cases be treated in terms of categorisation? Page 14 line 52: are the specific antibiotic names provided in brackets examples to the specific group of antibiotics? In other words, how will beta-lactam antibiotics other than cefuroxime (e.g., ceftazidime) or aminoglycosides other than gentamicin or tobramycin (e.g., amikacin) be treated in the analysis? How will the geographical distribution of AMR or isolates be categorized? Please briefly state. Page 9 lines 40 – 44: Since this meta-analysis will collectively exclude studies whereby patients with iterated specific conditions have been included, it may miss out some valuable data that can be extracted from such studies. Please consider reporting this as limitation.
--

REVIEWER	Soleimani , Mohammad Tehran University of Medical Sciences
REVIEW RETURNED	06-Jan-2023

GENERAL COMMENTS	Dear Authors, Many thanks; I think it may be beneficial to consider updated papers in 2022; at least these can be addressed: 1. Infectious keratitis: trends in microbiological and antibiotic sensitivity patterns. Soleimani M, Tabatabaei SA, Masoumi A, Mirshahi R, Ghahvechian H, Tayebi F, Momenaei B, Mahdizad Z, Mohammadi SS. Eye (Lond). 2021 Nov;35(11):3110-3115. doi: 10.1038/s41433-020-01378-w. Epub 2021 Jan 19. PMID: 33469134 Free PMC article. 2. A ten-year report of microbial keratitis in pediatric population under five years in a tertiary eye center. Soleimani M, Tabatabaei SA, Mohammadi SS, Valipour N, Mirzaei A. J Ophthalmic Inflamm Infect. 2020 Nov 27;10(1):35. doi: 10.1186/s12348-020-00227-x. PMID: 33245477 Free PMC article. 3. Fungal keratitis: An overview of clinical and laboratory aspects. Mahmoudi S, Masoomi A, Ahmadikia K, Tabatabaei SA, Soleimani M, Rezaie S, Ghahvechian H, Banafsheafshan A. Mycoses. 2018 Dec;61(12):916-930. doi: 10.1111/myc.12822. Epub 2018 Jul 27. PMID: 29992633 Review. Also, the readers may want to see this paper more organized, for example, by listing the preliminary sensitivity data in a table.
---

REVIEWER	Fernandes, Merle Visakhapatnam LV Prasad Eye Institute, Cornea
REVIEW RETURNED	10-Jan-2023

GENERAL COMMENTS	This is a well described methodology for a systematic review of AMR in microbial keratitis which will definitely be of use to clinicians. The major limitation would be the non availability of up to date break points for topical antibiotics in ophthalmic literature and hence conclusions would have to be drawn from existing break points from serum levels. In addition, there would be significant heterogeneity in the reporting formats. These have been addressed by the authors. It may also be useful to include data (wherever available), on the mutant prevention concentration since evidence may emerge on the trends of AMR for the different groups of antibiotics when comparing MIC 90 and MPC. The authors should include the culture positivity rate in their data.
--

VERSION 2 – AUTHOR RESPONSE

Reviewer: 1

Dr. Eray Atalay, Eskisehir Osmangazi University, Turkey

Comments to the Author:

This is a protocol paper describing a well-contemplated meta-analysis on the AMR in topical treatments for microbial keratitis. Considering the scope and breadth of the analysis, it will provide invaluable information both for clinicians and researchers alike. My comments can be found below.

Page 4 line 53: Please write the year range in full e.g., 2007 – 2009

This has been corrected (line 128).

Page 9 line 5: Please change the wording “recorded” to “record”. What do the authors mean by eye-appropriate topical antimicrobials. Please clarify.

We have corrected this typographic error (line 204).

We have included the following definition of ‘eye-appropriate antimicrobials’(line 205-207):

“We define eye-appropriate antimicrobials as agents that are either commercially available or specifically compounded for topical ophthalmic use.”

Page 9 line 22: Although it is understood in the preceding paragraph, please clarify whether “case series” or “case reports” will be included/excluded. Please also clarify whether studies that report keratitis of viral origin (with/without antiviral susceptibility testing) will be included/excluded. Although this has been partially addressed in response to a previous comment, the cited sentence creates confusion as to whether “cases” or “case series” of other isolates will be included or not. Please also report the target isolate groups (bacterial, fungal and amoebic) in the abstract.

We will include case series but not individual case reports as we require a representative sample of isolates from a particular setting with information on the proportion with AMR. The text has been modified:

“We will include case series but not individual case reports as we require a representative sample of isolates from a particular setting with information on the proportion with AMR.” (lines 215 and lines 223-225)

Studies that only report keratitis of viral origin will be excluded. We have added the following text to the Abstract:

“Studies that only report on viral keratitis will not be included.” (line 37).

This message is repeated in the section ‘Types of studies, eligibility, and setting’ (lines 215-216).

We have specified the target isolate groups in the Abstract with the following text:

“bacterial, fungal, or amoebic”, (line 36).

Page 9 line 42: please write acronyms (ICU, SJS) in full at first use.

This has been done (lines 226 and 227)

Page 11 line 49: there is emerging data that CNS can also demonstrate methicillin resistance. How will these cases be treated in terms of categorization?

We will be recording, and grouping, both the types of organisms as well as their MICs. If a study reports the proportion of MRSA (or MRCNS), we will record this first in the 'Bacterial groups' in the text (lines 275-283) as well as in the Data Extraction Table. If an MIC for methicillin is given, then we will group that result into the beta-lactam sensitivity data (see below).

We have not specified methicillin MICs as a data file because methicillin is a proxy agent for cephalosporin resistance in Staph aureus. In fact, methicillin susceptibility in S.aureus is an indicator of susceptibility to all cephalosporins, and all beta-lactamase stable, beta-lactams, i.e. oxacillin, flucloxacillin, co-amoxiclav, tazocin, meropenem, etc., many of which are 'eye appropriate'. The situation is further complicated because many laboratories have moved away from using methicillin as the proxy agent in favor of ceftazidime, which gives clearer separation of resistant/susceptible strains.

Page 14 line 52: are the specific antibiotic names provided in brackets examples to the specific group of antibiotics? In other words, how will beta-lactam antibiotics other than cefuroxime (e.g., ceftazidime) or aminoglycosides other than gentamicin or tobramycin (e.g., amikacin) be treated in the analysis?

Yes, the antibiotics named in brackets are examples of the specific groups chosen for analysis. We have added "e.g." to each bracket to clarify this. We have added the following text to clarify how we will handle data if more than one antibiotic in a class is tested against the same organisms: "If a study reports susceptibility for more than antimicrobial in a class against the same organism(s), our strategy for the primary analysis will be to select the results for the most frequently used antibiotic of that class with reference to the whole data set. We will then perform a sensitivity analysis substituting results for the less commonly used antibiotic(s) in that class to check that the results do not change substantially." (lines 334-339)

How will the geographical distribution of AMR or isolates be categorized? Please briefly state.

We will use the Global Burden of Disease regions classification. This information, with a reference, is contained in the section with details of the Meta-regression (349-350).

"Region: we will group studies according to the global super-region as defined by the Global Burden of Disease Study.[62]"

For clarity, we have also added this information to the Abstract:

"... between Global Burden of Disease (GBD) regions.." (line 46).

Page 9 lines 40 – 44: Since this meta-analysis will collectively exclude studies whereby patients with iterated specific conditions have been included, it may miss out some valuable data that can be extracted from such studies. Please consider reporting this as limitation.

We have stated in the Abstract that the review aims to provide global and regional prevalence estimates of antimicrobial resistance in corneal isolates. We then state that we will exclude studies that exclusively report from specific environments (e.g., Intensive Care) or from patients with specific conditions (e.g., Stevens Johnson syndrome). It is known that the spectrum of bacterial isolates from patients in ICU or from patients with SJS may differ from the regional prevalence data. Importantly, we have not said that we will exclude these patients or risk factors if they are included as part of a larger case series. We believe the manuscript is clear on this point and the text has not been changed.

Reviewer: 2

Dr. Mohammad Soleimani, Tehran University of Medical Sciences

Comments to the Author:

Dear Authors,

Many thanks; I think it may be beneficial to consider updated papers in 2022; at least these can be addressed:

Thank you. We will ensure that these publications are considered for inclusion in the systematic review

1.

Infectious keratitis: trends in microbiological and antibiotic sensitivity patterns.

Soleimani M, Tabatabaei SA, Masoumi A, Mirshahi R, Ghahvechian H, Tayebi F, Momenaei B, Mahdizad Z, Mohammadi SS.

Eye (Lond). 2021 Nov;35(11):3110-3115. doi: 10.1038/s41433-020-01378-w. Epub 2021 Jan 19.

PMID: 33469134 Free PMC article.

2.

A ten-year report of microbial keratitis in pediatric population under five years in a tertiary eye center.

Soleimani M, Tabatabaei SA, Mohammadi SS, Valipour N, Mirzaei A. J Ophthalmic Inflamm Infect. 2020 Nov 27;10(1):35. doi: 10.1186/s12348-020-00227-x.

PMID: 33245477 Free PMC article.

3.

Fungal keratitis: An overview of clinical and laboratory aspects.

Mahmoudi S, Masoomi A, Ahmadikia K, Tabatabaei SA, Soleimani M, Rezaie S, Ghahvechian H, Banafsheafshan A.

Mycoses. 2018 Dec;61(12):916-930. doi: 10.1111/myc.12822. Epub 2018 Jul 27.

PMID: 29992633 Review.

Also, the readers may want to see this paper more organized, for example, by listing the preliminary sensitivity data in a table.

We have organized the protocol according to the Author's Instructions for BMJ Open.

We have presented the rationale for performing this systematic review (lines 190-195). We do not think that the addition of a table of preliminary but incomplete data would be helpful.

Reviewer: 3

Dr. Merle Fernandes, Visakhapatnam LV Prasad Eye Institute

Comments to the Author:

This is a well described methodology for a systematic review of AMR in microbial keratitis which will definitely be of use to clinicians.

The major limitation would be the non-availability of up to date break points for topical antibiotics in ophthalmic literature and hence conclusions would have to be drawn from existing break points from serum levels. In addition, there would be significant heterogeneity in the reporting formats. These have been addressed by the authors.

We have mentioned in the Introduction that there are no established ophthalmic breakpoints for topically applied antimicrobials (lines 108-111). We have also mentioned that results must be extrapolated from serum levels for systemically administered antimicrobials (line 108-111), and we acknowledge the associated uncertainty in the interpretation of this data. However, this is more relevant to the comparison of clinical outcome as opposed to the comparison of susceptibility levels and MICs. The different criteria for reporting susceptibility have been acknowledged and

these criteria will be noted in the data form. However, because these definitions (e.g., EUCAST) are consistent, any change in AMR will be reported against this stable background. Importantly, we are not estimating the clinical relevance of the results.

It may also be useful to include data (wherever available), on the mutant prevention concentration since evidence may emerge on the trends of AMR for the different groups of antibiotics when comparing MIC 90 and MPC.

MPC is something not routinely measured and, in our preliminary survey of the literature, rarely, if at all reported in relevant ophthalmic papers. However, we have added this field to the Data Collection form.

The authors should include the culture positivity rate in their data.

Although this is not directly relevant to the goals of this study, this has been included as a field in the Data Collection form.

VERSION 3 – REVIEW

REVIEWER	Atalay, Eray Eskisehir Osmangazi University, Turkey
REVIEW RETURNED	03-Feb-2023
GENERAL COMMENTS	All necessary revisions have been made.